# Comparison of Two Efficient Methods for Calculating Partition Functions

**DOI:** 10.3390/e21111050

**Published:** 2019-10-28

**Authors:** Le-Cheng Gong, Bo-Yuan Ning, Tsu-Chien Weng, Xi-Jing Ning

**Affiliations:** 1Institute of Modern Physics, Fudan University, Shanghai 200433, China; 15110200001@fudan.edu.cn; 2Applied Ion Beam Physics Laboratory, Fudan University, Shanghai 200433, China; 3Center for High Pressure Science & Technology Advanced Research, Shanghai 202103, China; boyuan.ning@hpstar.ac.cn (B.-Y.N.); tsuchien.weng@hpstar.ac.cn (T.-C.W.)

**Keywords:** partition function, statistical physics, molecular dynamics simulations

## Abstract

In the long-time pursuit of the solution to calculating the partition function (or free energy) of condensed matter, Monte-Carlo-based nested sampling should be the state-of-the-art method, and very recently, we established a direct integral approach that works at least four orders faster. In present work, the above two methods were applied to solid argon at temperatures up to 300 K. The derived internal energy and pressure were compared with the molecular dynamics simulation as well as experimental measurements, showing that the calculation precision of our approach is about 10 times higher than that of the nested sampling method.

## 1. Introduction

The birth of statistical physics laid a solid foundation to predict thermodynamic properties of macroscopic condensed matters. Phase transitions [1,2,3], protein folding [4] and the optimal conditions for novel material growth could be predicted theoretically as long as the partition function (PF) or free energy can be evaluated [5]. Nevertheless, solutions to the PF have been a long standing problem [6] and attempts were reluctantly turned to the help of molecular simulations [7]. With precedent efforts made in calculating the relative free energy, e.g., Gibbs ensemble Monte Carlo (MC) [8] sampling and thermodynamic integration [9], more attentions have been paid to density of states (DOSs) for computing the absolute PF [10,11,12,13]. The Bayesian-statistics-based nested sampling (NS) may be the state-of-the-art one [14,15], which aims at uniformly sampling a series of fixed fractions partitioned by potential energies in the configurational space to calculate DOS and has been applied in several systems [1,16,17,18,19,20,21,22,23,24,25].

Very recently, we put forward a direct integral approach (DIA) to calculate the PF of condensed matters [26] and the high accuracy has been proved by molecular dynamics (MD) simulations of condensed copper using tight-binding potential [26], graphene and γ-graphyne materials using Brenner potential [27]. Based on our reinterpretation of the original sense of the integral, it was shown that DIA works at least four-order faster than NS [26]. On the other hand, it has not yet been confirmed whether the DIA has improved the computational precision of precedent MC methods. In this work, we carried out detailed analysis of DIA and NS in terms of the computational precision, and performed MD simulations to test the precision of internal energy and equations of state derived from the PF. It should be pointed out that the tests with MD simulations, instead of experimental data, are the most rigorous because same interatomic potentials can be used in calculations of the PF and MD simulations, which have been proved to be capable of producing very accurate results for various systems [28,29,30]. If the results derived from PF are only compared with experimental measurements, just as in most previous works, it would yet be doubted that the method for calculating the PF is accurate or not even if the agreements are excellent since it would be very likely that a deficient algorithm combined with an inappropriate empirical potential accidentally gives rise to an outcome close to the experiment.

The paper is organized as follows. In Section 2, NS and DIA were briefly formulated, and in Section 3, we first discussed the relationship between efficiency and accuracy of NS, and then performed MD simulations of solid argon to test the computational precision of DIA and NS, showing that DIA has a much higher precision than NS. In addition, we found that NS works badly for the highly-condensed systems while DIA has no such a problem. A comparison with experimental data of solid argon along the melting line was presented as well, which further validates that DIA is more accurate than NS.

## 2. Methods

PF is defined as a summation over the probabilities of all the microstates, and for a canonical ensemble consisting of N particles confined in volume *V* at temperature *T*, it reads
(1)$$Z\left(N,V,T\right)=\frac{1}{N!\Lambda^{3N}}\int dq^{3N}\exp\left[-\beta U\left(q^{3N}\right)\right],$$
where Λ is the thermal wavelength, $\beta =1/k_{B}T$ with $k_{B}$ the Boltzmann constant, $q^{3N}=\left\{q_{1},q_{2}\ldots ,q_{3N}\right\}$ the Cartesian coordinates of particles and $U(q^{3N})$ the potential energy. The 3N-dimensional integral on the right hand of Equation (Equation 1) is solely related to the microscopic states in configurational space, the so-called configurational integral (CI),
(2)$$Q=\int dq^{3N}\exp\left[-\beta U\left(q^{3N}\right)\right].$$

### 2.1. Nested Sampling

In Equation (Equation 2), microstates in configurational space are expressed in terms of coordinates of particles. From another point of view, we may also label the microstates by their corresponding potential energy and the integral can be rewritten in terms of the DOS of potential energy.

To introduce this, Equation (Equation 2) is re-expressed in reduced coordinates(s=q/L) as
(3)$$Q=V^{N}\int_{0}^{1}ds^{3N}\exp\left[-\beta U\left(s^{3N}\right)\right],$$
where *L* is the length of the cubic simulation box and *V* is the volume. The integral in Equation (Equation 3) represents the ensemble average of Boltzmann factors and denoted as $Q_{ex}$. Since the probability of finding potential energy *U* is proportional to the DOS, $\Omega \left(U\right)=\int_{0}^{1}\delta \left(U-U\left(s^{N}\right)\right)ds^{N}$, $Q_{ex}$ can be expressed as [16,20]
(4)$$Q_{ex}=\frac{\int \exp(-\beta U)\Omega (U)dU}{\int \Omega (U)dU}.$$

The strategy of NS is to partition the configurational space into a series of energy-decrease subdivisions numbered by *m*. For the *m*th subspace with upper energy limit $U_{m}$, a fixed number of configurations (*L*) with each energy $\varepsilon_{i}<U_{m}$ are generated by MC method and ordered in a sequence as $\varepsilon_{1}<\varepsilon_{2}<\ldots <\varepsilon_{L}$. The lower energy boundary $U_{m+1}$, which is the upper one for the (m+1)th subspace, is set to be the energy of a fixed fraction α of current subspace, as $U_{m+1}=\varepsilon_{I}$ with I=αL. By the NS algorithm, Equation (Equation 4) can be simplified as [22]
(5)$$Q_{ex}\approx \sum_{m}\frac{\int_{U_{m+1}}^{U_{m}}\Omega \left(U\right)dU}{\int \Omega (U)dU}\exp\left(-\beta {\langle U\rangle }_{m}\right)=\sum_{m}\omega_{m}\exp\left(-\beta {\langle U\rangle }_{m}\right),$$
where ${\langle U\rangle }_{m}$ is an averaged energy of the *m*th subspace and $\omega_{m}$ stands for the percentile of the *m*th phase space volume. It is obvious that $\omega_{m}=\alpha^{m}-\alpha^{m+1}$, and after *n*th iteration when the convergence condition is reached, CI is evaluated as
(6)$$Q=V^{N}Q_{ex}=V^{N}\sum_{m=1}^{n}\left(\alpha^{m}-\alpha^{m+1}\right)\exp\left(-\beta {\langle U\rangle }_{m}\right),$$
where ${\langle U\rangle }_{m}=\left(U_{m}+U_{m+1}\right)/2$ which is the average of two consecutive median energies of each sampled partition to approximate an averaged energy of the *m*th subspace [20,22]. According to $E=-\frac{\partial \ln Z}{\partial \beta }$, the internal energy of the N-particle system is calculated by
(7)$$\begin{array}{cc} E & =\frac{3}{2}Nk_{B}T-\frac{\partial Q_{ex}}{\partial \beta }\\ \; & =\frac{3}{2}Nk_{B}T+\frac{\sum_{m=1}^{n}\left[\left(U_{m}+U_{m+1}\right)/2\right]\left(\alpha^{m}-\alpha^{m+1}\right)\exp\left[-\beta \left(U_{m}+U_{m+1}\right)/2\right]}{\sum_{m=1}^{n}\left(\alpha^{m}-\alpha^{m+1}\right)\exp\left[-\beta \left(U_{m}+U_{m+1}\right)/2\right]}. \end{array}$$

For determining the pressure by $P=\frac{1}{\beta }\frac{\partial \ln Z}{\partial V}$, another CI for the system with a volume of V+ΔV should be calculated and *P* is obtained by
(8)$$\begin{array}{cc} P & =\frac{1}{\beta }\left(\frac{N}{V}+\frac{\partial \ln Q_{ex}}{\partial V}\right)\\ \; & \approx \frac{1}{\beta }\left(\frac{N}{V}+\frac{1}{Q_{ex}\left(V\right)}\frac{Q_{ex}\left(V+\Delta V\right)-Q_{ex}\left(V\right)}{\Delta V}\right). \end{array}$$

### 2.2. Direct Integral Approach

Consider Equation (Equation 2) and let the set $Q^{3N}=\left\{Q_{1},Q_{2}\ldots Q_{3N}\right\}$ be the coordinates of particles in the state of the lowest potential energy $U_{0}$, we may introduce a function as
(9)$$U^{\prime }\left(q^{\prime 3N}\right)=U\left(q^{3N}\right)-U_{0},$$
where $q_{i}^{\prime }=q_{i}-Q_{i}$. By inserting Equation (Equation 9) into Equation (Equation 2), we obtain
(10)$$Q=e^{-\beta U_{0}}\int dq^{\prime 3N}\exp\left[-\beta U^{\prime }\left(q^{\prime 3N}\right)\right].$$

According to our very recent work [26], the integral can be solved as
(11)$$Q=e^{-\beta U_{0}}\prod_{i=1}^{3N}L_{i},$$
where $L_{i}$ represents the effective length on the *i*th degree of freedom and is defined as
(12)$$L_{i}=\int e^{-\beta U^{\prime }\left(0\ldots q_{i}^{\prime }\ldots 0\right)}dq_{i}^{\prime }.$$

For homogeneous systems with certain geometric symmetry, such as perfect one-component crystals, all the particles are equivalent and $U^{\prime }$ felt by one particle moving along $q_{x}^{\prime }$ may be the same as the one along $q_{y}^{\prime }$ (or $q_{z}^{\prime }$). In such a case, Equation (Equation 11) turns into
(13)$$Q=e^{-\beta U_{0}}L^{3N},$$
where L is determined by Equation (Equation 12). Otherwise, it is needed to calculate the effective length, $L_{x}$, $L_{y}$, $L_{z}$ respectively, and Equation (Equation 11) turns into
(14)$$Q=e^{-\beta U_{0}}{\left(L_{x}L_{y}L_{z}\right)}^{N},$$
and, *E* and *P* are thus evaluated as
(15)$$\begin{array}{ccc} E & = & \frac{3}{2}Nk_{B}T+U_{0}+3N\frac{\sum_{i=1}^{n}U_{i}\exp\left[-\beta U_{i}\right]}{\sum_{i=1}^{n}\exp\left[-\beta U_{i}\right]}, \end{array}$$
(16)$$\begin{array}{ccc} P & \approx & -\frac{U_{0}\left(V+\Delta V\right)-U_{0}\left(V\right)}{\Delta V}+\frac{3N}{\beta }\frac{1}{L(V)}\frac{L(V+\Delta V)-L(V)}{\Delta V}. \end{array}$$

## 3. Comparisons and Discussion

The tested models are face-centered-cubic (FCC) solid Ar systems consisting of 500 or 4000 atoms confined in a cubic box with different sizes, and, NS and DIA were applied to calculate internal energy *E* and pressure *P* at different temperatures to be compared with MD simulations. The interatomic potential for solid Ar was characterized by the commonly used pairwise 12-6 Lennard-Jones (L-J) potential [31],
(17)$$\phi \left(r_{ij}\right)=4\epsilon \left[{\left(\frac{\sigma }{r_{ij}}\right)}^{12}-{\left(\frac{\sigma }{r_{ij}}\right)}^{6}\right],$$
where $r_{ij}$ is the distance between atoms *i* and *j*, ϵ=117.05 (K), σ=3.4 Å and the cutoff distance is $r_{cut}=12.0$ Å. The MD simulations with periodic boundary condition applied were performed by the Large-scale Atomic/Molecular Massively Parallel Simulator software package [32] with a time step of 0.1 fs. The Nose-Hoover constant-temperature algorithm [33] was used to produce a canonical ensemble at temperature *T*. The pressure was computed by the virial theorem for two-body potentials [34,35], the algorithm of which has already been integrated in the software package. The system was allowed to relax 20 ps at first and then continued to run for another 50 ps, during which averages of *E* and *P* were recorded in every 10 fs.

To implement NS, it should be at first to select appropriate values of α and *L*, which cooperatively balance the computational efficiency and precision of Q [Equation (Equation 6)]. Apparently, the larger the values of α and *L* are, the higher the calculation precision is, but the slower the computation speed is. Although the initial choice of α made by Pártay et al. [16] is L/(L+1), successive works [17,18,21,22] have showed that a smaller value of α=1/2 is sufficient enough to guarantee the calculation precision and enables NS to be applicable to systems consisting of up to several hundred atoms, of which the computational cost is too expensive for NS with α=L/(L+1). Therefore, α=1/2 was adopted in this work. Cares should be also paid to the value of *L* because, besides the factors of efficiency and systematic errors mentioned above, fluctuations of the calculated results in NS simulations closely depend on *L* for a fixed α [22]. We performed NS with four different numbers of configuration (L= 45,000, 60,000, 75,000, 90,000) to calculate the pressures of the solid Ar system consisting of 500 atoms with a density of 1.83 g/cm${}^{3}$ at different temperatures, where the well-built cage model for solid systems [18] was used. For each *L*, we ran the NS simulations 15 times to produce the averaged value of pressure which was compared with MD simulations to see the relationship between the deviations and *L*.

As shown in Figure 1 (data in Appendix A), the pressures obtained by the NS are gradually approaching to those of the MD simulations as *L* increases and the corresponding fluctuations of NS is relatively larger with the smallest *L*. On the other hand, it should be noted that the fluctuations does not monotonically decrease with the increase of *L*, which was also observed in previous works [17]. The fluctuations for L= 60,000 and 90,000 are almost the same, which are about 30% smaller than those with L= 75,000, though the pressures with L= 90,000 are slightly closer to the MD simulations. Considering that the computational time with L= 90,000 is twice as much as that with L= 60,000, we chose L= 60,000 in the following work and conducted the NS simulations at each (N,V,T) conditions for 15 times to calculate the averaged values of internal energy by Equation (Equation 7) and pressure by Equation (Equation 8), where the volume difference ΔV was made by changing the length of the box by 1% because our calculations showed that smaller volume difference would produce very unphysical results.

Relatively, systematic parameters are much fewer for implementation of DIA. For the solid Ar system, the atoms were placed right at the FCC sites to produce $U_{0}$, and $U^{\prime }\left(0\ldots q_{i}^{\prime }\ldots 0\right)$ in Equation (Equation 12) was obtained by moving the center atoms along its *Z*-axis ([100] direction) by 2Å while the coordinates of its *X*-axis, *Y*-axis, and of all the other particles were kept fixed. $2\times 10^{4}$ potential energies were recorded to calculate the L by Equation (Equation 12), and, the internal energy and pressure were subsequently calculated by Equations (Equation 15) and (16), where the volume difference was made by changing the length of the box by $10^{-3}\%$.

For the argon system of 500 atoms with different densities (1.83, 2.13, 2.43 and 2.98 g/cm${}^{3}$) at temperatures from 25 K to 300 K, *E* and *P* obtained by DIA and the NS are shown in Figure 2, where the corresponding quantities of $E_{MD}$ and $P_{MD}$ of MD simulations are also presented as comparisons (data in Appendix A). For the systems with a density of 1.83 g/cm${}^{3}$ and 2.13 g/cm${}^{3}$, the averaged relative difference of internal energy, RDE ($=|\frac{E-E_{MD}}{E_{MD}}|$), of DIA is less than 4.1%, which is about four times smaller than that, 16.6%, of NS [36]. As the density increases up to 2.43 g/cm${}^{3}$ and 2.98 g/cm${}^{3}$, the averaged RDE of DIA decrease to 5.51% and 0.48% respectively, while the averaged RDE of the NS climbs up to 36.44% for the density of 2.43 g/cm${}^{3}$ and the NS fails to work for the system with density of 2.98 g/cm${}^{3}$. As to precision of the pressure, the averaged relative difference, RDP ($=|\frac{P-P_{MD}}{P_{MD}}|$), of DIA is 2.48%, 1.69%, 0.17% and 0.63% for the densities of 1.83, 2.13, 2.43 and 2.98 g/cm${}^{3}$ respectively, while the corresponding RDP of the NS is 10.22%, 9.18%, 4.54% and *∞*.

The above comparisons show that the calculation precision of DIA is much higher than that of the NS. Furthermore, DIA works better with increase of the density while the NS can hardly work when the density is higher than 2.98 g/cm${}^{3}$. The difficulty should be attributed to numerical calculations of Equation (Equation 6), where the factor ($\alpha^{m}-\alpha^{m+1}$) approaches to zero (α=1/2) as *m* approaches to larger number, meanwhile, the factor $e^{-\beta (U_{m}+U_{m+1})/2}$ increases quickly when $U_{m}<0$, which is the common case for the Ar systems with lower density and the product ($\left(\alpha^{m}-\alpha^{m+1}\right)\cdot e^{-\beta (U_{m}+U_{m+1})/2}$) is not too large (or small) for the 16 bit number of computer to describe. However, when the density is large enough that the $U_{m}>0$, both ($\alpha^{m}-\alpha^{m+1}$) and ($e^{-\beta (U_{m}+U_{m+1})/2}$) approach to zero as *m* getting larger, and the product ($\left(\alpha^{m}-\alpha^{m+1}\right)\cdot e^{-\beta (U_{m}+U_{m+1})/2}$) gets to be so small (but not exactly “0”) that the output of computer is exact “0”, which makes the denominator in Equation (Equation 7) be zero easily. For this reason, we failed to apply the NS to calculate *E* and *P* of the Ar system with a density of 2.98 g/cm${}^{3}$. A larger value of α might be helpful while the computational efficiency would be slowed down. By contrast, DIA has no such a problem because the largest part of the potential energy, $U_{0}$ of the MSS, has been extracted in Equations (Equation 9) and (Equation 10), and the left part $U^{\prime }$ is small enough to guarantee the precision of the integral for high density systems.

The lower precision for the NS calculating the pressure can be understood as follows. The pressure is determined by Equation (Equation 8), where the volume difference ΔV should be set as small as possible to achieve high precision. However, the integral Q of Equation (Equation 6) is not very sensitive to the small changes of the volume *V* because of the random characteristic of MC simulations, leading to large fluctuations of Q(V+ΔV)−Q(V) for each running of the MC simulation. Our calculations showed that the large fluctuations would produce unphysical pressures when the ΔV is smaller than $10^{-4}\%V$, which corresponds to the length of the cubic box changed by 1% adopted in our calculations. In DIA for calculating the pressure [Equation (16)], the involved quantities $U_{0}$ and L determined by Equation (Equation 12) are all sensitive to volume of the system, so the volume difference in Equation (16) can be set much smaller. We tried several values of the box length difference in the range of $10^{-1}\%$–$10^{-6}\%$ and confirmed that the obtained pressures converges at the volume difference of $10^{-13}\%$ ($10^{-3}\%$ box length difference).

The computational efficiency of the NS and DIA depends on the number of the total potential calculation. For the NS running, the MC algorithm has to work $6\times 10^{3}$–$8\times 10^{3}$ times each producing 60,000 configurations to reach the convergence, so $3.6\times 10^{8}$–$4.8\times 10^{8}$ times of potential energy calculations must be performed to produce the $U_{m}$ in Equation (Equation 6). Because of the fluctuations, the NS was run 15 times for a given system to produce the averaged results, thus the number of the total potential calculations is larger than $5.4\times 10^{9}$, which is about five orders of magnitude larger than the one, $2\times 10^{4}$, for running DIA in the same system.

Because of the ultra-high efficiency, DIA was applied to calculate the internal energy and pressure of solid argon composed of 4000 atoms, on which the NS costs too much computer hours and we have to give up the calculations, and we performed MD simulations to give comparisons. As shown in Figure 3, both *E* and *P* obtained by DIA coincide well with MD simulations where both RDE and RDP of DIA are almost the same as those calculated in the 500-atom system (data in Appendix A).

Finally, a comparison was made of DIA and the NS with experimental data of solid Ar along melting line [37]. Considering the lower efficiency of the NS, the simulated system for both DIA and NS consists of 500 atoms and the computational procedures are the same as described above. As shown in Figure 4, the internal energy and pressure obtained by DIA are significantly better than those of the NS. The averaged relative deviation of internal energy and pressure to the experimental data is 5.34% and 5.5% for DIA, which are about 6 times smaller than the ones, 39.12% and 28.72%, for the NS.

## 4. Conclusions

In summary, by comparisons with MD simulations as well as experimental data, we confirmed that the accuracy of DIA outperforms the NS. The precision of DIA is about four times higher than that of NS for low-density systems and about one order higher in high-density situations. We also analyzed the intrinsic deficiency of NS in calculations of systems under highly condensed situations. Since the efficiency of DIA is at least five orders faster than that of the NS at the same time, DIA paves a better way to investigate thermodynamic properties of condensed matters, especially the ones with high density under extreme conditions.

## Figures and Tables

**Figure 1 entropy-21-01050-f001:**
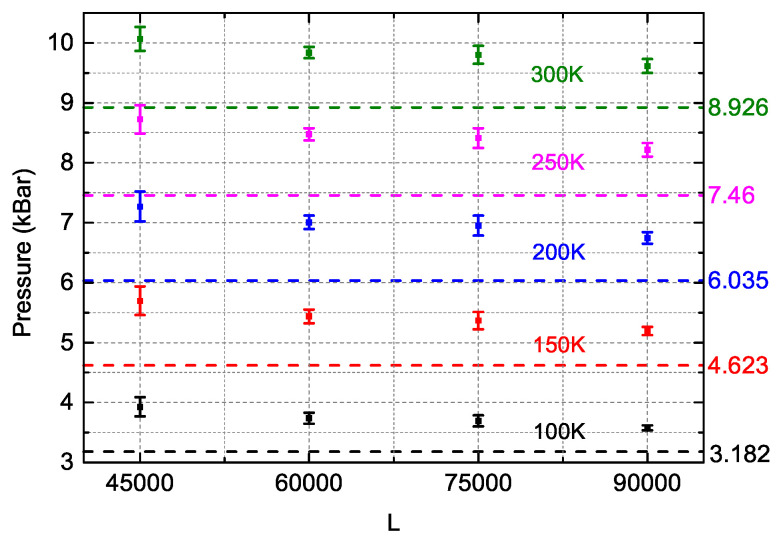
The dependence of the pressure at different temperatures obtained by the NS (Nested Sampling) and the standard deviations upon the *L*, where the results of MD (Molecular dynamics) simulations are illustrated in dashed lines.

**Figure 2 entropy-21-01050-f002:**
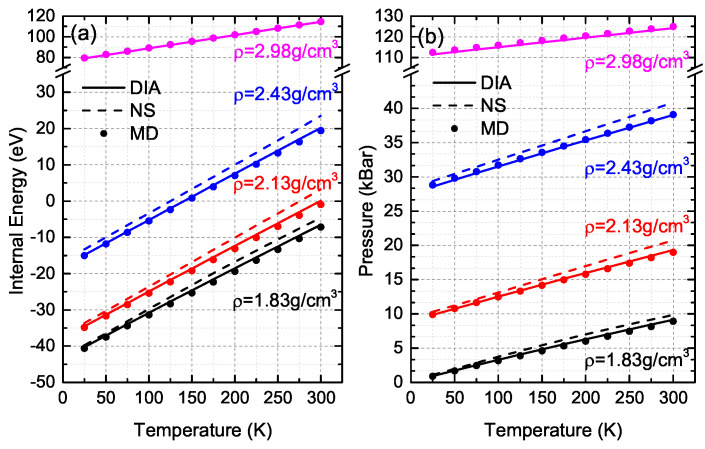
Internal energy (**a**) and pressure (**b**) of 500 argon atoms in the solid state obtained by DIA (Direct Integral Approach) (solid line), the NS (dashed line) and MD simulations (circles). Different color stands for different density ρ.

**Figure 3 entropy-21-01050-f003:**
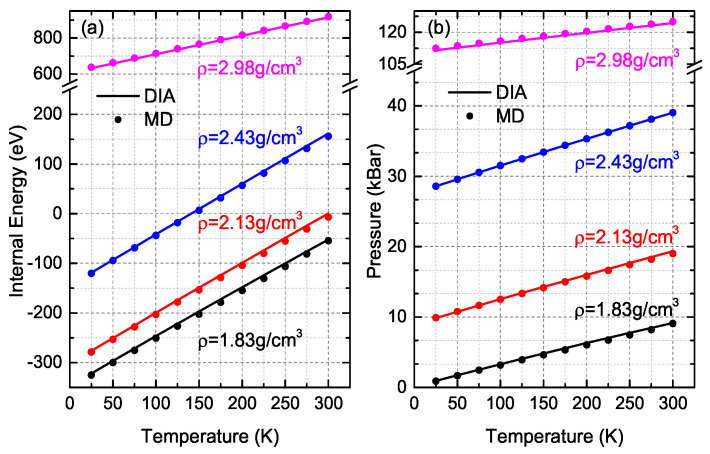
Internal energy (**a**) and pressure (**b**) of 4000 argon atoms in the solid state obtained by DIA (solid line) and MD simulations (circles). Different color stands for different density ρ.

**Figure 4 entropy-21-01050-f004:**
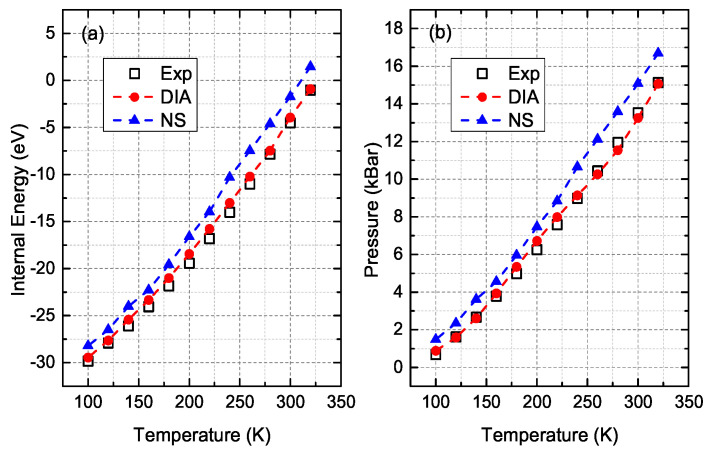
Internal energy (**a**) and pressure (**b**) of solid-state Ar, from experiment (squares) [37], DIA (red circles), NS (blue triangles) along the melting line.

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
