# Peer review of "Comparison of Two Efficient Methods for Calculating Partition Functions"

_entropy, 2019, doi:10.3390/e21111050_

Round 1
Reviewer 1 Report
The Authors present in "Comparison of two efficient methods for calculating partition functions" a comparison of basic thermodynamic observables E and p (internal energy; pressure, respectively) to be calculated from the statistical-thermodynamical formalism based on the partition function(s) of a canonical (T=const.) ensemble.
The method is rather employed to calculate E and p for an argon-atom system consisting of N atoms, wherein N is finite. The approximations made to achieve the goal are simple mathematical finite-difference based calculations, and a deep rationale associated with it is clearly missing.
The obtained (E,p) doubled formulae for NS/nested sampling, Eqs. (6) and (7), and comparatively for DIA/direct integral approach, Eqs. (14) and (15) are, in fact, hardly comparable because they do rest on different physical quantities.
As a matter of fact, also the title of the ms. reviewed is misleading since the final output of two methods employed is always the (E,p) couple, reflected by Figures 2-4, but not the partition function which possesses a unique and only one definition in the Gibbs statistical mechanics, namely that represented by Eq. (1) of the ms.
Also the language used in the paper is a bit obsolete and/or imprecise; for example, directly after Eq. (5) as an approximation the Authors used two-point arithmetic mean not the average, the latter name being here reserved for the ensemble average.
All in all, the ms. is not suitable form in its present form and is judged by the present reviewer to undergo major methodological and nomenclature concerning revision before being submitted again to the Journal.
Moreover, the quality of Figure 4, especially the lines drawn within it, are not acceptable, except that they can be described as quasi-guiding lines or so.
As a drawback to be removed, if agreed, a plausible quantum effect in the tightly packed system of argon atoms can be commented, as suggested by a paper titled "Polymorphic phase transitions in systems evolving in a two-dimensional discrete space" Phys. Rev. E 60, 1252 (1999), to be virtually involved in the literature, if useful.
To sum up, the paper is not recommended for publication in its present form, and it ought to undergo a substantial, thus major revision, especially along the comparison lines clearly embodied in the pairs of formulae: (6) and (14) as well as (7) and (15) in order to convince the reader thoroughly on which formal substantiation a subtle difference is going to rely.
Author Response
Dear reviewer,
We are sincerely grateful to you for the careful review and suggestive comments for the manuscript. The replies and relevant revisions to the manuscript are listed as follows:
As emphasized in the review, the partition function (PF) is indeed a unique value by its definition, and both nested sampling (NS) and direct integral approach (DIA) do aim at calculating the same PF. Following your point of view, we found that a term, VN, was accidentally missing in Eq.(3), which may cause trouble to elucidate the scheme of NS. We are deeply sorry for this mistake.
The term, VN and corresponding terms have been corrected in Eqs. (3), (6) and (8), and more details of the mathematical derivations of NS and related statements were added in the current manuscript. We again apologize for such a mistake.
The obsolete and imprecise language has been corrected according to your comments.
As to the quality of Figure 4, we added circles and triangles for DIA and NS simulations respectively and the dash lines are for the guide to the eye as suggested.
We thank for the suggested reference and has added it as Ref. [3].
Thanks for your reviewing and we are looking forward to your further valuable comments.
Reviewer 2 Report
The manuscript presents two computational methods namely Nested sampling (NS) approach and Direct integral approach (DIA) to estimate partition functions and other thermodynamic properties. The accuracy is validated against the results from atomistic molecular dynamics simulations. The system considered in this study is condensed phase of argon (in face centered cubic lattice) where the interactions are described using L-J type pairwise potential. There are many interesting results presented in this work. Firstly, The authors show that with increasing number of L (number of different configurations), the calculated pressures in NS approach become comparable to those obtained from MD simulations. The study also compares the accuracy of NS and DIA in computing partition functions and properties such as energies and pressures. It shows that the DIA provides comparably accurate results when compared to NS approach. Further it is shown that the latter approach does not work for higher densities. The authors also performed a study with relatively larger size where DIA is shown to work and provide results for E and P comparable to that of MD simulations. This work is suitable for publication in “Entropy” journal. I recommend its publication once the following remarks are addressed. No further review is needed.
(i) Even though it is easy to understand how the energy of a condensed phase system is estimated in MD simulations, it is not clear how the pressure has been computed. May be authors can provide the expression used to compute the pressure in this study.
(ii) This is also not clear to me how the energies are computed experimentally. In particular, I am referring to Figure 4.
(iii) This is only a suggestion. Wherever it is relevant, you can make the codes/software available for readers to reproduce the results presented in the manuscript or to use.
Author Response
Dear reviewer,
We are sincerely grateful to you for the careful review and suggestive comments for the manuscript. The replies and relevant revisions to the manuscript are listed as follows:
The detail about how the pressure was computed in this study has been added in the current manuscript.
The detail about how the energies are computed experimentally could be seen in Ref. [38].
Thanks for your reviewing and we are looking forward to your further valuable comments.
Round 2
Reviewer 1 Report
The Authors have very well provided the suggested corrections, and even more, they have creatively introduced a necessary mathematical corrections, they have accurately pointed out in red in their revised ms. Figure 4 is also technically well corrected.